# Luteolin Protects Pancreatic β Cells against Apoptosis through Regulation of Autophagy and ROS Clearance

**DOI:** 10.3390/ph16070975

**Published:** 2023-07-07

**Authors:** Ming Han, Yuting Lu, Yunhua Tao, Xinwen Zhang, Chengqiu Dai, Bingqian Zhang, Honghong Xu, Jingya Li

**Affiliations:** 1School of Chinese Materia Medica, Nanjing University of Chinese Medicine, Nanjing 210046, China; 2State Key Laboratory of Drug Research, The National Center for Drug Screening, Shanghai Institute of Materia Medica, Chinese Academy of Sciences, Shanghai 201203, China; 3University of Chinese Academy of Sciences, Beijing 100049, China; 4School of Pharmaceutical Science and Technology, Hangzhou Institute for Advanced Study, University of Chinese Academy of Sciences, Hangzhou 310024, China

**Keywords:** luteolin, type 2 diabetes, β cell, autophagy, reactive oxygen species

## Abstract

Diabetes, which is mainly characterized by increased apoptosis and dysfunction of beta (β) cells, is a metabolic disease caused by impairment of pancreatic islet function. Previous studies have demonstrated that death-associated protein kinase-related apoptosis-inducing kinase-2 (*Drak2*) is involved in regulating β cell survival. Since natural products have multiple targets and often are multifunctional, making them promising compounds for the treatment of diabetes, we identified *Drak2* inhibitors from a natural product library. Among the identified products, luteolin, a flavonoid, was found to be the most effective compound. In vitro, luteolin effectively alleviated palmitate (PA)-induced apoptosis of β cells and PA-induced impairment of primary islet function. In vivo, luteolin showed a tendency to lower blood glucose levels. It also alleviated STZ-induced apoptosis of β cells and metabolic disruption in mice. This function of luteolin partially relied on *Drak2* inhibition. Furthermore, luteolin was also found to effectively relieve oxidative stress and promote autophagy in β cells, possibly improving β cell function and slowing the progression of diabetes. In conclusion, our findings show the promising effect of *Drak2* inhibitors in relieving diabetes and offer a potential therapeutic target for the protection of β cells. We also reveal some of the underlying mechanisms of luteolin’s cytoprotective function.

## 1. Introduction

Diabetes is a chronic metabolic disease. Type 1 diabetes (T1D) and type 2 diabetes (T2D) patients account for 95% of patients with diabetes. Beta (β) cell apoptosis is the principal process leading to diabetes. T1D is caused by autoimmunity, and T-cell-mediated β cell destruction is nonpathogenic [1]. The attenuation of β cells is always accompanied by the development of T2D [2]. The decreased β cell number is mainly caused by apoptosis, as the proliferation of β cells is not different between T2D patients and normal individuals [3]. Many factors, such as oxidation, glucotoxicity, lipotoxicity, and endoplasmic reticulum stress (ER stress), trigger β cell apoptosis [4]. Clinically, antidiabetic drugs used now exhibit excellent hypoglycemic effects and other beneficial effects, but other than glucagon-like peptide 1 (GLP-1)-related drugs, which have beneficial effects on β cells, these drugs cannot ameliorate β cell apoptosis or loss. Moreover, clinically used antidiabetic drugs have side effects, such as weight gain, hypoglycemia, gastrointestinal disease, and increase cardiovascular risk. Therefore, it is necessary to identify novel therapeutic targets for diabetes and develop effective hypoglycemic drugs that protect against β cell apoptosis, improve islet function, and have fewer side effects.

Death-associated protein kinase-related apoptosis-inducing kinase-2 (*Drak2*) is a member of the death-associated protein kinase (DAPK) family, and all five members of this family participate in cell apoptosis [5,6,7,8,9]. *Drak2* is highly expressed in lymphoid tissue. It is mainly related to immunoregulation and influences β cell survival [9]. The islets of *Drak2* transgenic mice were found to be more susceptible to free fatty acid (FFA)-induced apoptosis and glucose-stimulated insulin secretion (GSIS)-mediated damage, while si*Drak2* was shown to reverse this damage [10]. Therefore, *Drak2* may be a potential target for diabetes treatment. In addition, *Drak2* has been reported to be involved in the pathogenesis of other metabolic diseases, such as nonalcoholic fatty liver disease, and the mechanism by which *Drak2* affects mitochondrial function through alternative splicing has been elucidated. Therefore, as a possible target for the treatment of diabetes, it is particularly important to elucidate the mechanism underlying its role in diabetes.

Unlike existing drugs, natural products have relatively few side effects. Natural plant products have exhibited excellent efficacy in the treatment of various diseases for a long time. Hyperglycemic chemicals extracted from plants usually exhibit multiple therapeutic effects, including antioxidant, anticancer, anti-inflammatory, and antidiabetic effects, rather than single effect. There are already some natural products used in antidiabetic research, such as berberine, resveratrol, and so on.

As an important pathogenic factor in the development of diabetes, high glycolipid toxicity triggers oxidative stress in the body by promoting the production of free radicals and leads to the occurrence and development of diabetes and its complications [11]. However, pancreatic β cells exhibit low levels and activity of the intracellular antioxidant enzymes superoxide dismutase (SOD), catalase (CAT), and plasma glutathione peroxidase (GSH-Px) [12] and are very sensitive to ROS-mediated damage. In addition to directly damaging β cells, ROS can also indirectly damage β cells by impacting signal transduction pathways that affect insulin synthesis and secretion. Therefore, reducing oxidative stress is a beneficial strategy for improving the quality of pancreatic β cells.

Flavonoids are widely distributed phytochemicals in dietary plants and Chinese herbal medicines. Many studies have found that most flavonoids exhibit excellent antioxidant activity, and some dietary flavonoids show promising antidiabetic and hypoglycemic effects as well [13]. This suggests that natural products may be of great benefit for the protection of pancreatic islet β cells. Luteolin, as a natural product belonging to the group of dietary flavonoids, also has antioxidant effects [14]. A previous study demonstrated that luteolin could reduce the formation of nitric oxide (NO) and inducible nitric oxide synthase (iNOS). It also upregulated the expression of the transcription factor MafA in β cells, thus increasing the secretion of insulin in urea-damaged β cells [15], which proved that luteolin may also protect islet β cells via its antioxidant activity. Autophagy is an important cellular process. When cells lack energy or are exposed to environmental stresses, they can produce a double membrane structure, recognize substrates, form vesicles, and fuse with lysosomes to remove aging organelles and degrade proteins [16]. Autophagy homeostasis is crucial to the function of β cells, and β cell damage is closely related to a decrease in autophagy. Autophagy activation was found to be insufficient to meet proteolytic requirements in T2D (*db/db, ob/ob*) mice [17,18]. Knockout of autophagy-related gene 7 (*Atg7*) leads to hyperglycemia, glucose tolerance impairment, and mitochondrial damage [19,20]. It has been reported that prolonged exposure of β cells to FFAs, such as palmitate (PA), leads to an increased number of autophagosomes due to blocked autophagic flux, which is clearly indicated by increased microtubule-associated protein light chain 3 II (LC3-II) levels as well as ubiquitin-binding receptor p62 accumulation [21,22]. Impairment of autophagy turnover in β cells leads to apoptosis cell death. The promotion of autophagy directly increases insulin secretion [23]. Autophagy contributes to β cell survival through multiple pathways and plays diverse roles in the survival and function of β cells. Therefore, the promotion of autophagy may enhance GSIS and protect β cells from apoptosis when autophagy is impaired by PA. These studies suggest that the function of β cells may be improved through autophagy regulation.

In this study, we identified luteolin as a *Drak2* inhibitor through screening of a natural product library and found that luteolin could effectively promote β cell autophagy, prevent β cell lipotoxicity-induced apoptosis, and slightly ameliorate the symptoms of T2D mice. We generated mice with conditional β cell-specific *Drak2* knockout (*Drak2^f/f^:Cre*) and proved that luteolin exerted its effect partly through *Drak2* inhibition. Furthermore, luteolin reduced PA-induced increases in ROS levels in a dose-dependent manner, which might have contributed to protection against β cell apoptosis. The molecular mechanism of luteolin in autophagy regulation and β cell protection was partially clarified. We hope this work will provide a theoretical basis for the development of new antidiabetic agents. 

## 2. Results

### 2.1. Luteolin Acts as a Drak2 Inhibitor

Our previous work validated that *Drak2* inhibitors can prevent pancreatic β cells from undergoing lipotoxicity-induced cell apoptosis [24]. Therefore, we established a kinase activity detection assay in vitro using the ADP-Glo^TM^ Kinase Assay kit (Promega, Madison, WI, USA) and screened the Natural Product and Derivatives Focus Library to identify small molecules that can inhibit *Drak2* activity. The natural products we screened are listed in Appendix A. ADP-Glo^TM^ Kinase Assay in vitro showed that some of these compounds affected *Drak2* activity with more than 50% inhibitory rate at the primary screened concentration of 20 μM. Among them, luteolin showed the strongest effect with an IC_50_ of 346.7 ± 30.0 nM (Figure 1a,b). Compound 22b was reported to be an effective *Drak2* inhibitor, so it was used as the positive control in this study [13]. As mentioned above, luteolin also has been reported to have antidiabetic effects. Therefore, we assumed it was worth further investigating the mechanism underlying the association between its inhibitory effect on *Drak2* and islet β cell protection.

### 2.2. Luteolin Protects β Cells from Lipotoxicity

To confirm whether luteolin protects β cells in vitro, the apoptosis rate and glucose-stimulated insulin secretion (GSIS) were measured in INS-1E cells after PA and luteolin cotreatment for 24 h. The cell apoptosis rate increased markedly to 55.79% in INS-1E cells after PA treatment, while the apoptosis rate decreased to 36.71% after treatment of 22b (5 μM) (Figure 2a). The values of apoptosis rate mentioned in this section are the average of three independent experiments. As Figure 2a shows, luteolin alleviated cell apoptosis in a dose-dependent manner, and the apoptosis rate declined from 48.09% (2.5 μM) to 44.42% (5 μM) and then to 35.83% (10 μM) (Figure 2a). Western blot analysis showed that the levels of PARP, cleaved caspase3 and cleaved caspase9, as apoptosis marker proteins, were significantly increased at 24 h after PA treatment, and cotreatment with luteolin and PA reduced the apoptosis rate (Figure 2b). In terms of β cell function, the GSIS function of primary islets was impaired after PA treatment, while luteolin prevented PA-induced impairment of GSIS function of mouse primary islets in a dose-dependent manner (Figure 2c). Analysis of the level of lactate dehydrogenase (LDH), which can indicate cytotoxicity, showed that 22b and luteolin were safe (Appendix A).

These results illustrated that luteolin acted as a *Drak2* inhibitor and alleviated PA-induced cell apoptosis and GSIS impairment in a dose-dependent manner.

### 2.3. Luteolin Protection of β Cell Function Is Partially Dependent on *Drak2* Inhibition

To prove that the effect of luteolin is related to *Drak2*, we overexpressed *Drak2* in INS-1E cells, and the levels of apoptosis marker proteins increased (Figure 3a). INS-1E cells were more sensitive to PA-induced damage after *Drak2* overexpression. Luteolin treatment dose-dependently alleviated the damage caused by *Drak2* overexpression (Figure 3a and Appendix A), and luteolin mostly reversed the impairment of GSIS (Figure 3b).

To illustrate the correlation between luteolin and *Drak2*, we knocked down *Drak2* in INS-1E cells and then performed a fluorescence-activated cell sorting (FACS). The apoptosis rate increased from 27.98% (control group) to 37.51% after PA treatment, but 5 μM and 10 μM luteolin reduced it to 23.74% and 17.99%, respectively (Figure 3c). When *Drak2* was knocked down in INS-1E cells, the apoptosis rate was reduced to 16.20%, whereas it was 27.98% in the si-control group, and PA treatment only increased the apoptosis rate to 18.99%. This indicates that *Drak2* may participate in β cell apoptosis regulation. Nevertheless, after treatment with luteolin, the apoptosis rate was further lowered to 11.54% (5 μM) and 13.11% (10 μM), which suggests that luteolin can protect β cells from apoptosis, but this effect may not be entirely dependent on its inhibitory effect on *Drak2* (Figure 3c). We harvested samples treated with si*Drak2* and 0.4 mM PA for 24 h to detect apoptosis marker protein levels. Similar to the FACS results, *Drak2* knockdown weakened the effect of luteolin, which proved our assumption (Figure 3d and Appendix A). A GSIS assay was performed after the same treatment (Figure 3e). The effects of luteolin were weakened by si*Drak2*. The results above demonstrated that luteolin could maintain β cell function by inhibiting *Drak2*.

To confirm the dependence of the effect of luteolin on *Drak2*, *Drak2*-deficient islets (isolated from *Drak2^f/f^:Cre* mice) were used to validate whether luteolin still affects GSIS capability. As expected, neither 22b nor luteolin could further increase GSIS by *Drak2*-deficient islets (Figure 3f). The level of LDH in these same samples was measured, and the results demonstrated that insulin secretion was not caused by cytotoxicity (Appendix A).

Through overexpression or knockdown of *Drak2*, we confirmed that luteolin improved β cell function partly through *Drak2* inhibition.

### 2.4. Luteolin Treatment Reduces ROS Production and Protects β Cells from Oxidative Stress

Oxidative stress promotes the progression of T2D pathogenesis [4]. Luteolin is a flavonoid compound. It can be inferred from its chemical structure that luteolin can react with oxygen radicals (Figure 1a). To determine whether luteolin protects β cell function in vitro via antioxidant activity, we measured the ROS level in INS-1E cells after *Drak2* overexpression and PA treatment. Intercellular ROS levels increased after *Drak2* overexpression and were further increased in the presence of PA (Figure 3f). Vitamin C (VC), as a positive control, showed excellent ROS clearance ability, while luteolin decreased ROS levels in a dose-dependent manner (Figure 3f). This result suggested that luteolin could resist oxidative stress caused by *Drak2* overexpression and PA treatment. When we overexpressed superoxide dismutase (SOD or MnSOD), intercellular ROS levels decreased significantly in INS-1E cells. Overall, the higher the transfection efficiency was, the lower the ROS content (Appendix A). SOD decreased *Drak2* protein levels under the same conditions (Appendix A). In brief, these results suggest that the effects of luteolin are partly related to decreasing oxidative stress.

### 2.5. Luteolin Alleviates Cell Apoptosis Partially through Autophagy Promotion

According to a previous study on luteolin, luteolin promotes autophagy in cancers and inflammation [25,26]. We hypothesized that luteolin may also prevent cytotoxicity via autophagy flux promotion in β cells.

To prove our assumption, the autophagy inhibitors 3-methyladenine (3-MA) and chloroquine (CQ) were used to illustrate the mechanism by which luteolin improves β cell survival and function. The FACS results showed that 3-MA and CQ both aggravated cell apoptosis, resulting in apoptosis rates of 44.64% and 51.83%, respectively; these increases in the apoptosis rate were mostly reversed by luteolin in PA-treated INS-1E cells, reducing the apoptosis rate to 34.44% and 35.54% (Figure 4a). Autophagy was blocked by CQ and 3-MA. The levels of marker proteins of autophagy and apoptosis were significantly increased. The expression of autophagy marker proteins further increased after luteolin treatment. The above results prove that luteolin can effectively promote autophagic flux (Figure 4b and Appendix A). Regarding β cell function, 3-MA and CQ significantly impaired primary islet GSIS, while luteolin almost completely reversed this impairment (Figure 4c). 3-MA and CQ showed no cytotoxicity during treatment (Appendix A). Western blot analysis showed that autophagy was suppressed after CQ treatment in the presence of PA, and the levels of autophagy marker proteins were increased. However, marker proteins further accumulated after luteolin treatment, which meant that autophagic flux was promoted by luteolin (Figure 4d and Appendix A).

In summary, these results indeed prove that luteolin exerts cytoprotective effects by promoting autophagy.

### 2.6. Luteolin Can Promote Insulin Secretion and Alleviate Diabetes Process In Vivo

In vitro experiments have already demonstrated the cytoprotective effect of luteolin, and we were interested in its function in vivo. First, we performed a tissue distribution assay with Institute of Cancer Research (ICR) mice. Luteolin was mainly distributed in the pancreas after i.p. injection (Figure 5a) but scarcely distributed to the pancreas after oral delivery (Appendix A). The pharmacokinetic parameters of luteolin after p.o. and i.v. administration in ICR mice indicated its poor oral bioavailability and rapid elimination (Appendix A and Appendix A). Based on these findings, we evaluated the efficacy of luteolin when delivered by i.p. injection. The oral glucose tolerance test (OGTT) was performed after luteolin treatment, and TAK-875, a G protein-coupled receptor 40 (GPR40) agonist, was used as a positive control. The OGTT results showed that luteolin as well as TAK-875 markedly improved glucose clearance (Figure 5b).

Additionally, we established a T2D mouse model by administration of a low dosage of streptozotocin (STZ). Metformin, a common and effective hypoglycemic drug [27], was used as a positive control in the experiment. Littermates were used as nondiabetic controls. Body weight and food intake were monitored every three days, and there were no differences among the groups (Appendix A). Postprandial blood glucose (PBG) levels were measured every 3 days, and fasting blood glucose (FBG) levels were measured every 9 days. Metformin showed an excellent ability to improve PBG levels from day 39 to the end of the experiment (Figure 5c). Luteolin did not show a distinct hypoglycemic effect during 42 days of administration, but it showed a slight hypoglycemic effect after 18 days of administration. On Day 27, FBG levels were significantly decreased in the 20 mg/kg luteolin treatment group (Figure 5c). After 42 days of treatment, analysis of aspartate transaminase (AST), alanine transaminase (ALT), and triglyceride (TG) levels in plasma showed that luteolin could improve metabolic indices in mice (Figure 5d). According to insulin and glucagon immunofluorescence staining in pancreatic sections, luteolin slightly increased the number and area of β cells in islets (Figure 5e). Hematoxylin and eosin (H&E) staining of the pancreas showed similar results (Figure 5f). 

Luteolin promoted GSIS after a single i.p. injection in ICR mice. Long-term hypo-glycemia and islet dysfunction was induced by STZ, but luteolin did not have a marked effect in controlling blood glucose levels.

## 3. Discussion

Considering the potential role of *Drak2* in the development of diabetes, we screened a natural product library for compounds that inhibit *Drak2* activity and identified luteolin as a lead compound. Luteolin ameliorated PA-induced apoptosis and primary islet function impairment in vitro. In vivo experiments showed that a single administration of luteolin could lead to acute hypoglycemia. Multiple administrations could alleviate the pathogenesis of STZ-induced diabetes and alleviate β cell apoptosis. In addition, we verified that the effect of luteolin was partially dependent on *Drak2* inhibition and found that luteolin could also exert its effect by relieving oxidative stress and promoting autophagy.

As the name implies, *Drak2* is primarily associated with apoptosis, similarly in pancreatic β cells. Several studies have shown that *Drak2* plays an important role in the development of type 1 diabetes. The mRNA and protein levels of *Drak2* in pancreatic β cells are rapidly increased in response to inflammatory lymphokine stimulation, ultimately leading to apoptosis of islet β cells [28]. Similarly, *Drak2* overexpression was found to promote apoptosis in islet β cells treated with FFA [10], but the mechanism by which FFA cause *Drak2* overexpression was not elucidated. Therefore, *Drak2* may serve as a novel target for the development of drugs for diabetes. This study determined that targeted inhibition of *Drak2* activity can effectively alleviate apoptosis in β cells. Studies have reported that natural product inhibitors of *Drak2* can alleviate cell apoptosis and improve GSIS in vitro [24].

Natural plant products are a very large potential resource for drug discovery. There are already some natural products used in antidiabetic research, such as berberine and resveratrol. We hoped to identify compounds that alleviate β apoptosis by inhibiting *Drak2* activity with fewer side effects than currently used antidiabetic drugs. Therefore, a natural product library was screened, and luteolin was identified as an effective natural product. The current study found that luteolin has multiple activities. Luteolin, a polyphenolic bioflavonoid, exists in multiple Chinese traditional medicines. Luteolin (20 µM) was found to inhibit the growth of glioblastoma cells [25], exhibiting antitumor activity. Luteolin influences the expression of anti-inflammatory cytokines such as IL-37 and IL-38 during an inflammatory process [29]. The doses of luteolin commonly administered to cells and animals are 100 µM and 20 or 50 mg/kg, respectively. Moreover, luteolin can exert antidiabetic effects and improve insulin secretion. Luteolin was used at 100 mg/kg [30] to treat diabetes in mice [31]. 

Given that luteolin is a flavonoid, we also compared the antidiabetic mechanism of luteolin with that of other flavonoid natural products. Flavonoids mainly exert hypoglycemic effects through the following mechanisms: (1) enhancing insulin secretion, reducing apoptosis of pancreatic β cells and promoting the proliferation of β cells; (2) regulating key enzymes involved in glucose metabolism; (3) regulating the expression of proteins related to insulin signaling pathways and enhancing insulin sensitivity; (4) inhibiting inflammation and oxidative stress; and (5) improving lipid metabolism and reducing lipid toxicity. For example, the flavonol compound quercetin can improve GSIS, inhibit the expression of oxidative stress-related genes and iNOS, and inhibit the translocation of nuclear factor (NF-κB) and the release of cytochrome C, thereby preventing apoptosis of β cells [32]. The mechanism by which anthocyanins regulate glycolipid metabolism mainly involves inhibiting the enzymatic activities of peroxisome proliferator-activated receptor gamma coactivator-1 alpha (PCG-1α), phosphoenolpyruvate carboxykinase (PEPCK) and glucose 6 phosphatase (G-6-Pase); inhibiting hepatic gluconeogenesis and insulin resistance; regulating lipid metabolism; and inhibiting lipid accumulation and oxidation [33]. The isoflavone puerarin exerts hypoglycemic effects by upregulating the expression of insulin receptor substrate-1 (IRS-1), insulin-like growth factor-1 (IGF-1) and peroxisome proliferator-activated receptor-alpha (PPARα) [34]. Therefore, the role of luteolin in protecting β cells from apoptosis is consistent with the effects of other compounds with similar structures. We discovered another specific mechanism by which luteolin protects β cells from apoptosis that partially depended on inhibiting *Drak2*, which is a cell apoptosis promotion kinase.

Considering the close relationship between oxidative stress and diabetes, the effectiveness of antioxidant strategies in the treatment of diabetes has received increasing attention. Oxidative stress is a crucial factor in diabetes. Luteolin is a polyphenolic bioflavonoid. It contains multiple phenolic hydroxyls, which implies that it may exhibit reducibility. Studies have shown that luteolin can resist oxidation. The doses of luteolin administered to cells and animals were 5 or 10 μM and 20 mg/kg, respectively [35]. We demonstrate that luteolin reduces ROS levels in cells to protect them from PA-induced apoptosis. Recently, studies have also shown that luteolin alleviates epithelial-mesenchymal transformation induced by oxidative injury in ARPE-19 cells via the Nrf2 and AKT/GSK-3β pathway [15]. There are also studies showing that luteolin alleviates cognitive impairment in an Alzheimer’s disease mouse model by inhibiting endoplasmic reticulum stress-dependent neuroinflammation [36]. Numerous studies have shown that luteolin can effectively relieve damage caused by oxidative stress in different diseases. In addition, several studies have revealed the correlation between ROS and *Drak2* levels. *Drak2* responds to ROS in cells to regulate apoptosis. ROS can regulate *Drak2* through PKD during T-cell activation [37]. Similarly, in our study, overexpression of SOD could resist ROS production to reduce the level of *Drak2*. This may be another cause of *Drak2* downregulation. However, in our study, the signaling pathways involved in luteolin’s specific antioxidant effect in pancreatic β cells were not studied in detail, and our future research will further explore the mechanism of this aspect.

Autophagy is closely related to β cell function and survival. Various factors (such as high levels of glucose or fatty acids) can enhance β cell autophagy to varying degrees, resulting in further removal of toxic proteins such as amylin from β cells, thus ensuring insulin synthesis and secretion. Some studies have revealed the mechanism underlying the effect of luteolin on autophagy. The dose of luteolin administered to cells to regulate autophagy was 5 or 10 μM [25,26]. The protective effect of luteolin against diabetic cardiomyopathy in rats is related to its ability to reverse JNK-suppressed autophagy [38]. Studies have shown that luteolin can affect other processes involved in autophagy. Luteolin inhibits autophagy in allergic asthma by activating PI3K/Akt/mTOR signaling and inhibiting the Beclin-1-PI3KC3 complex [39]. All the above results prove that luteolin can participate in the regulation of autophagy. We also obtained an interesting result in our study. The effect of luteolin is associated with *Drak2*, and other members of the DAPK family, to which DRAK2 belongs, have been reported to be associated with autophagy. According to previous reports, the main members of the DAPK family (DAPK1, DAPK2, DAPK3) are involved in autophagy regulation [40,41,42,43,44]. DAPK1 regulates the dissociation of beclin 1 from its Bcl-2 inhibitors. It is essential for the autophagic activity of beclin 1. DAPK2 directly interacts with and phosphorylates mTORC1 and participates in suppressing mTOR activity to promote autophagy induction. Therefore, does the effect of luteolin on autophagy also depend on the inhibitory activity of *Drak2*? The detailed relationship between *Drak2* and autophagy needs to be further studied.

In this study, we revealed the correlation among luteolin, autophagy, and *Drak2* and validated that luteolin can promote autophagy, thus improving islet β cell survival and function. Our work provides a potential strategy for the treatment of diabetes. However, there are many remaining questions related to luteolin, such as the effective dose and its solubility. We look forward to developing luteolin as an oral medicine through further research.

## 4. Materials and Methods

### 4.1. Source of Compounds

The natural products and derivatives library including luteolin for original screening was provided by Professor Jianmin Yue from Shanghai Institute of Materia Medica, Chinese Academy of Sciences. For the cellular and animal experiments, luteolin (HY-N0162) was purchased in powder form from Med Chem Express (MCE). Positive control 22b was provided by Professor Jie Tang from East China Normal University, the detailed synthesis method was as described previously [24]. TAK-875 (HY-10480), metformin (HY-B0627) and streptozotocin (HY-13753) were purchased from MCE. PA (112-39-0), CQ (50-63-5) and 3-MA (5142-23-4) were purchased from Sigma-Aldrich.

### 4.2. ADP-Glo Kinase Assay

ADP-Glo kits (Promega, Madison, WI; #V6903) were used for *Drak2* inhibitor screening. *Drak2* activity was measured and calculated from the amount of ADP generated from the enzyme reaction. The compounds were dissolved in DMSO and diluted to the indicated concentration for the screening assay. *Drak2* and substrate ATP were diluted in 1X Buffer (HEPES 50 mM (pH 7.0), NaN_3_ 0.02%, Orthovanadate 0.1 mM, 5 mM MgCl_2_, 0.01% (*w/v*) bovine serum albumin). The 5 µL kinase reaction was performed using 1X kinase buffer that contained 1 μL of compound, 2 μL of *Drak2* (160 nM) and 2 μL of substrate ATP (20 μM), and incubated for 2 h at room temperature. Then, 2.5 μL of ADP-Glo™ Reagent was added to each reaction and incubated for 2 h at room temperature to deplete the remaining ATP. Finally, 5 μL of Kinase Detection Reagent was added to each reaction to convert ADP to ATP via the luciferase reaction, which was detected using an EnVision multilabel plate reader. 22b was used as a positive control in the assay.

### 4.3. Cell Culture and Plasmid Transfection

The INS-1E (rat insulinoma) cell line was generously donated by Dr. Yong Liu (Wuhan University). It was cultured in RPMI 1640 (Corning; #1868882) with 10% FBS (Gibco) at 37 °C, 5% CO_2_ and 21% O_2_ in a regular incubator. INS-1E cells before passage 30 were used. The pCEP4-HA-*Drak2* plasmid was kindly provided by Dr. Jiangping Wu (Centre Hospitalier de l’Université de Montréal). Cells were transfected with the desired plasmid according to manufacturer’s instructions and incubated for 48 h. In detail, INS-1E cells were transfected with Lipofectamine 2000 (Invitrogen, Carlsbad, CA, USA; #11668019) containing the basic vector or required plasmids in Opti-MEM (Invitrogen; #31985070). After 6 h of incubation, the medium was changed to RPMI 1640 with 10% FBS (Gibco; #10099141). The cells were incubated for 2 days to perform the following experiments.

### 4.4. Small Interfering RNA (siRNA) Transfection

INS-1E cells were cultured in petri dishes overnight. The cells were transfected with Lipofectamine 2000 (Invitrogen, Carlsbad, CA, USA; #11668019) containing the scramble siRNA sequences or *Drak2* siRNA sequences in Opti-MEM (Invitrogen; #31985070). After 6 h of incubation, the medium was changed to RPMI 1640 with 10% FBS (Gibco; #10099141), and these cells were incubated for another 2 days. Compounds were added to the medium 24 h before the flow cytometric analysis (FACS). After FACS, Western blot analyses were performed. The small interfering RNA sequences are supplied in the supplementary information. All the experimental supplies in this assay were RNase free.

### 4.5. Isolation and Culture of Primary Pancreatic Islets 

Mouse pancreatic islets were isolated using the Liberase digestion method [45]. Briefly, as described earlier [24], the digestion solution (2 mL HBSS containing 20 mM HEPES and 1 mg/mL collagenase P (Roche) enzyme solution) was injected into the common bile duct. After pancreas completely inflated, it was harvested to a 15 mL tube containing 2 mL cold collagenase P enzyme solution and then incubated for 20 min in a 37 °C water bath. The pellet was washed with 5 mL quenching buffer (HBSS containing 10% FBS) and then centrifuged at RT 300 g for 3 min. The supernatant was then removed. The centrifugal product was washed with 10 mL of cold quenching buffer and centrifuged again at RT 300 g for 3 min. The pellet was resuspended in 2 mL of 25% Ficoll, and 1 mL each of the next three layers was added separately at 23%, 21% and 13% to form the gradient. The Ficoll gradient was centrifuged at 2500 g for 20 min. The islet layer should be visible between the 13% and 21% gradient. The islets were isolated and washed twice with 5 mL HBSS buffer. These islets were then cultured overnight in RPMI 1640 containing 10% FBS for next step of the experiment.

### 4.6. Treatment of Islets and INS-1E with Palmitate

The isolated islets were cultured in 48-well plates with about 10 islets/well in RPMI 1640 with 10% FBS. The INS-1E cells were cultured in 48-well plates in RPMI 1640 with 10% FBS. Palmitate (Sigma-Aldrich, St. Louis, MO, USA; final concentration 0.4 mM or 0.25 mM; #P9767) was added to each well for 24 h to induce islets dysfunction and apoptosis as indicated in each experiment.

### 4.7. Glucose-Stimulated Insulin Secretion and LDH Measurement

Glucose stimulated insulin secretion (GSIS) was determined as previously described in detail [46]. Islets were cultured for 2 days in complete RPMI 1640 medium with 10% FBS in the absence or presence of various stimulants. Then 10 islets/well were selected and transferred to 48-well plates. The islets were then incubated in Krebs–Ringer bicarbonate HEPES buffer (KRBH, 135 mM NaCl, 3.6 mM KCl, 0.5 mM NaH_2_PO_4_, 1.5 mM CaCl_2_, 2 mM NaHCO_3_, 10 mM HEPES and 0.1 % BSA, pH 7.4) and 0.1% fatty acid-free BSA for 1 h and then incubated in KRBH buffer containing 2.8 mM or 16.7 mM glucose for another hour at 37 °C. One hundred microliters of supernatant was removed for determination of insulin levels. The insulin content was detected via a HTRF insulin assay kit (Cisbio, 62INSPEC). Cell membrane integrity, which is indicated by LDH, was measured using a CytoTox-ONE™ Homogeneous Membrane Integrity Assay kit (Promega, Madison, WI; #G7891) according to the manufacturer’s instructions.

### 4.8. Antibodies and Immunoblotting

Western immunoblotting was performed as described previously [47]. In brief, cells were lysed, sonicated and boiled at 100 °C for 10 min in sample buffer (50 mM Tris-HCl, 2% *w/v* SDS, 10% glycerol, 1% β-mercaptoethanol, 0.01% bromophenyl blue, pH 6.8). The cell lysates were separated by SDS-PAGE and transferred to nitrocellulose (NC) filter membranes. The membranes were first incubated with blocking buffer (TBS with 0.01% Tween 20 and 5% non-fat milk) for 1 h at room temperature and then incubated overnight at 4 °C in buffer containing the primary antibodies. The membranes were washed three times and then incubated with secondary antibodies for 1 h. After three washes, immunostaining was visualized using an electrochemiluminescence and ChemiDoc imaging system (Bio-Rad, Hercules, CA, USA). Anti-*Drak2* (#2294), anti-PARP (#9542), anti-cleaved caspase 3 (#9664), anti-cleaved caspase 9 (#9509), anti-ULK1 (#8054), anti-insulin (#3014), and anti-GAPDH were purchased from Cell Signaling Technologies (Danvers, MA, USA). Anti-LC3B antibody (#L7543) was purchased from Sigma-Aldrich (St. Louis, MO, USA). Anti-p62 antibody (#sc-25575) was purchased from Santa Cruz Biotechnology. 

### 4.9. Flow Cytometric Analysis (FACS)

For the cell flow cytometric analysis (FACS), cells were trypsinized, and the single-cell suspensions were then stained with annexin V and propidium iodide (KeyGEN BioTECH; #KGA108) in binding buffer for 15 min at room temperature. The cells were then analyzed with a Guava Flow Cytometer (Millipore, St. Charles, MO, USA; Beckman, IN, USA). The data were collected with FlowJo software 10.6.2.

### 4.10. Measurement of Intracellular ROS Production

The production of intracellular ROS was measured using the ROS detection reagents (Invitrogen) according to the manufacturer’s instructions. In brief, INS-1E cells were plated into 96-well plates. After treatment with 0.4 mM PA for 24 h, the cells were washed twice with PBS and supplied with phenol red-free RPMI-1640 containing 10 μM DCFH-DA dye and then incubated for 30 min at 37 °C in the dark. The cells were then washed with PBS three times and the DCF fluorescence intensity was measured using microplate fluorescence reader (excitation wavelength: 488 nm and emission wavelength: 530 nm).

### 4.11. Generation of Mice with Deletion of Drak2 in β Cells

Mice in which exon 4 of the *Drak2* allele was flanked with *loxP* sites (denoted *Drak2^f/f^* mice) were generated at the Shanghai Research Center for Model Organisms. To generate β cell-specific *Drak2* knockout mice (denoted *Drak2^f/f^:Cre* mice), *RIP-Cre* transgenic mice (*B6.Cg-Tg(Ins2-cre)25Mgn/J*, denoted *Cre* mice), which express Cre recombinase under the control of the rat insulin promoter, were intercrossed with *Drak2^f/f^* mice to generate *Drak2^flox/+^:Cre* mice. The *Drak2^flox/+^:Cre* mice were then bred with *Drak2^flox/+^* mice to generate *Drak2^f/f^:Cre* along with *Drak2^f/f^* and *Cre* mice. The mice were maintained under a 12 h light/dark cycle with free access to a normal chow diet (Shanghai Laboratory Animal Co., Ltd., Shanghai, China) and water at an accredited animal facility at the Shanghai Institute of Materia Medica. All the experimental procedures and protocols were approved by the Institutional Animal Care and Use Committees at the Shanghai Institute of Materia Medica.

### 4.12. Pharmacokinetics (PK) and Tissue Distribution of Luteolin

Luteolin was dissolved in DMSO (1/1 = *v*/*v*) for storage. A comparative pharmacokinetic study was performed in ICR mice (n = 12, male) weighing 25–28 g. The study was performed as per the Institutional Animal Care and Use Committees at the Shanghai Institute of Materia Medica. The dissolvent formulations used in the study were: (1) Group A (p.o.): DMSO/0.9% NaCl = 5/95 = *v*/*v*; and (2) Group B (i.v.): DMSO/Tween-80/0.5% HPMC = 10/10/80 = *v*/*v*/*v*. The orbital blood samples were collected after fasting for at least 12 h. Water was not restricted before administration. Then the mice were treated with luteolin. The administration volume of each group was 10 mL/kg and the administration dosages were: (1) 20 mg/kg (p.o.); and (2) 5 mg/kg (i.v.). The food was given 2 h after administration. Each group was divided into 2 groups to obtain blood samples separately. Orbital blood (100 μL) was collected at 3 min, 10 min, 15 min, 45 min, 2 h, 4 h, 8 h, and 24 h. The blood samples were added to an EDTA anticoagulant tube in ice within 30 min of collection and then centrifuged at 12,000 rpm for 2 min to separate the plasma. The supernatant was collected and stored at −80 °C. The samples were analyzed by Servier Laboratory. 

To test the tissue distribution of luteolin, we ordered 8 ICR mice weighing 25–28 g from Shanghai SLAC laboratory animals Co., Ltd. (Shanghai, China) The mice were randomly divided into 2 groups. The solvent of luteolin was DMSO, Tween 80 and 0.5% HPMC at a ratio of 1:1:18. Thirty minutes after intraperitoneal injection and oral administration of luteolin, the heart, brain, spleen, liver, pancreas, kidney, abdominal fat and blood of the mice were taken. Then the blood samples were centrifuged at 12,000 rpm for 2 min, and the supernatant was transferred to new tubes. The subsequent tests were performed by Servier Laboratories (Suresnes, France).

### 4.13. Animals 

We ordered 7-week-old C57 mice from Shanghai SLAC laboratory animals Co., Ltd. Low-dose STZ (50 mg/kg) was administered for 5 consecutive days. One week later, the mice were randomly divided into 4 groups according to their postprandial blood glucose, body weight, fasting blood glucose and fasting body weight, named as the vehicle group, metformin group, and luteolin group. The non-treated C57 mice were the model control group. The solvent of luteolin was DMSO, Tween 80 and 0.5%HPMC at a ratio of 1:1:18. In the vehicle group, 10 mL/kg of solvent was injected intraperitoneally every day. Metformin was orally administered at 250 mg/kg per day, and luteolin was injected intraperitoneally at 20 mg/kg per day.

### 4.14. Physiological Measurements

Food consumption was measured for individually caged mice by weighing food daily before the dark cycle for two days, and body weight was measured at the same time. Blood glucose was determined from tail vein blood before and after 6 h fasting using a glucometer (ACCU-CHEK Perfor, Roche, Rotkreuz, Switzerland). 

### 4.15. Glucose Tolerance Test (GTT) 

For the GTT, blood glucose was determined from tail vein blood using a glucometer (ACCU-CHEK Perfor, Roche, Rotkreuz, Switzerland) at the indicated interval before and after an oral dose of glucose. For fasting blood glucose (FBG) measurement, the *ICR* mice were fasted for 12 h (overnight) and then an oral dose of 3 g/kg glucose of body weight was administered for postprandial blood glucose measurement. 

### 4.16. Immunostaining of Pancreas Sections

The pancreas was fixed in 4% paraformaldehyde, embedded in paraffin, and cut into consecutive 5 μm sections. For immunofluorescence staining, fixed pancreatic sections were heated for 10 min in boiling citrate buffer (0.01 M, pH 6.0) for antigen retrieval. Subsequently, the sections were probed with primary antibodies, followed by incubation with specific secondary antibodies conjugated to Alexa 488 or 594 (Invitrogen, Carlsbad, CA, USA). Hoechst (Sigma-Aldrich, St. Louis, MO, USA; final concentration, 10 g/mL; #14530) was used to visualize the nucleus. The sections were analyzed by fluorescence microscopy (Olympus, Tokyo, Japan). Images of the islets were acquired and analyzed using the Image Pro Plus Software 6.0.

### 4.17. Statistical Analysis 

Student’s *t*-tests were performed, and *p* < 0.05 was considered statistically significant. For the animal and in vitro studies, data were presented as the mean ± standard error of the mean (SEM). An unpaired, two-tailed Student’s *t*-test was used for two-group comparisons. One-way ANOVA analysis was used for comparisons among multiple groups, which was followed by LSD post hoc test for the difference between any two groups. The asterisks are presented in the figures to indicate statistical significance as follows: ** p* < 0.05; ** *p* < 0.01; and *** *p* < 0.001. All graphs were plotted with GraphPad Prism software 8.0.2.

## 5. Conclusions

Our previous study suggested that *Drak2* inhibitors could prevent β cells from undergoing lipotoxicity-induced cell apoptosis. By screening the natural product and derivatives library, we found that luteolin, as a flavonoid, serves as an effective *Drak2* inhibitor. Our experiments validated that luteolin could protect β cell function through alleviating PA-induced β cell apoptosis and preventing GSIS impairment in a dose-dependent manner, which may have been partially related to its effect in inhibiting *Drak2* activity, reducing oxidative stress and promoting β cell autophagy. Our in vivo study showed that the single-dose administration of luteolin promoted GSIS and reduced hyperglycemia, while long-term administration improved metabolic indices and ameliorated islet dysfunction in a T2D mouse model.

In summary, luteolin, as a natural *Drak2* inhibitor, has no significant cytotoxicity. It can improve islet β cell dysfunction via autophagy promotion and its antioxidative effect, which is worthy of further study. However, our study has some limitations. First, the mechanism by which luteolin regulates autophagy through *Drak2* inhibition requires further research. Second, there are many remaining questions related to luteolin, such as the effective dose and its solubility, and these questions cannot be ignored. We look forward to developing derivatives of luteolin as an oral therapeutic agent through further research.

## Figures and Tables

**Figure 1 pharmaceuticals-16-00975-f001:**
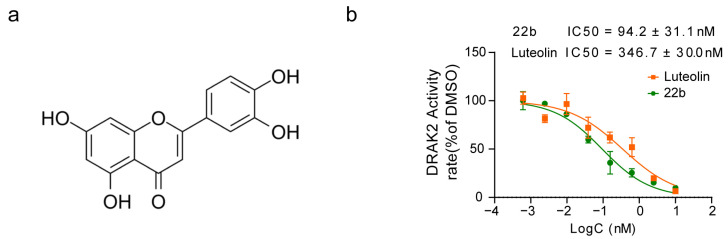
Structure of luteolin and molecular activity. (**a**) Chemical structure of luteolin (C_15_H_10_O_6_, Mw: 286.23). (**b**) Inhibitory activity of luteolin on *Drak2* was measured by ADP-Glo kinase assay, n = 4.

**Figure 2 pharmaceuticals-16-00975-f002:**
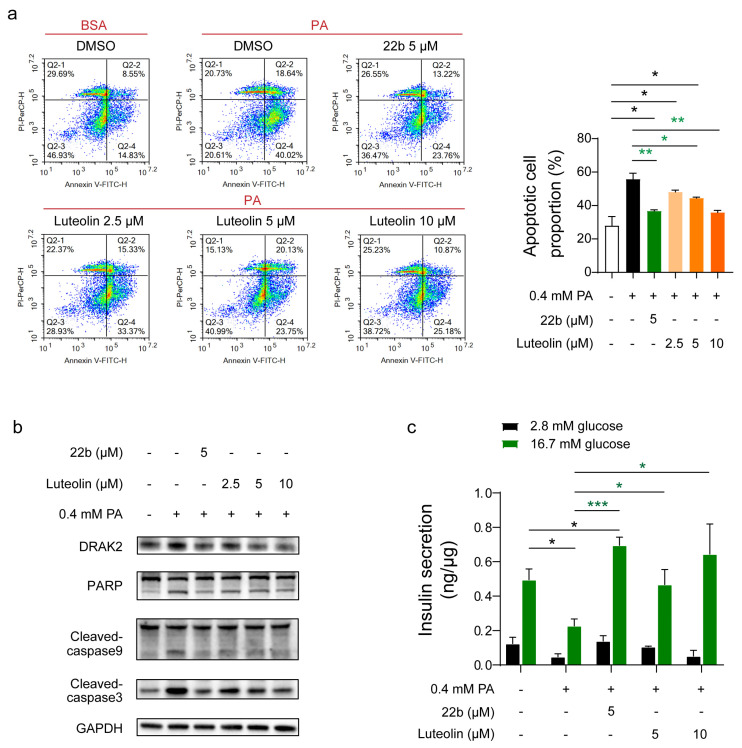
Effects of luteolin on inhibition of apoptosis and promotion of GSIS. (**a**) Representative FACS analysis and quantification of apoptotic rate with PA induced in response to treatments in INS-1E cells. Cell apoptosis was assessed by flow cytometry using Annexin V/PI staining in INS-1E cells, n = 3. (**b**) Apoptosis marker proteins PARP, Cleaved-caspase 3 and Cleaved-caspase 9 levels in INS-1E; 22b is a classical *Drak2* inhibitor and thus serves as a positive control. (**c**) GSIS analysis on primary islet, n = 3. All values are presented as mean ± SEM of at least three independent experiments; * *p* < 0.05, ** *p* < 0.01, *** *p* < 0.001.

**Figure 3 pharmaceuticals-16-00975-f003:**
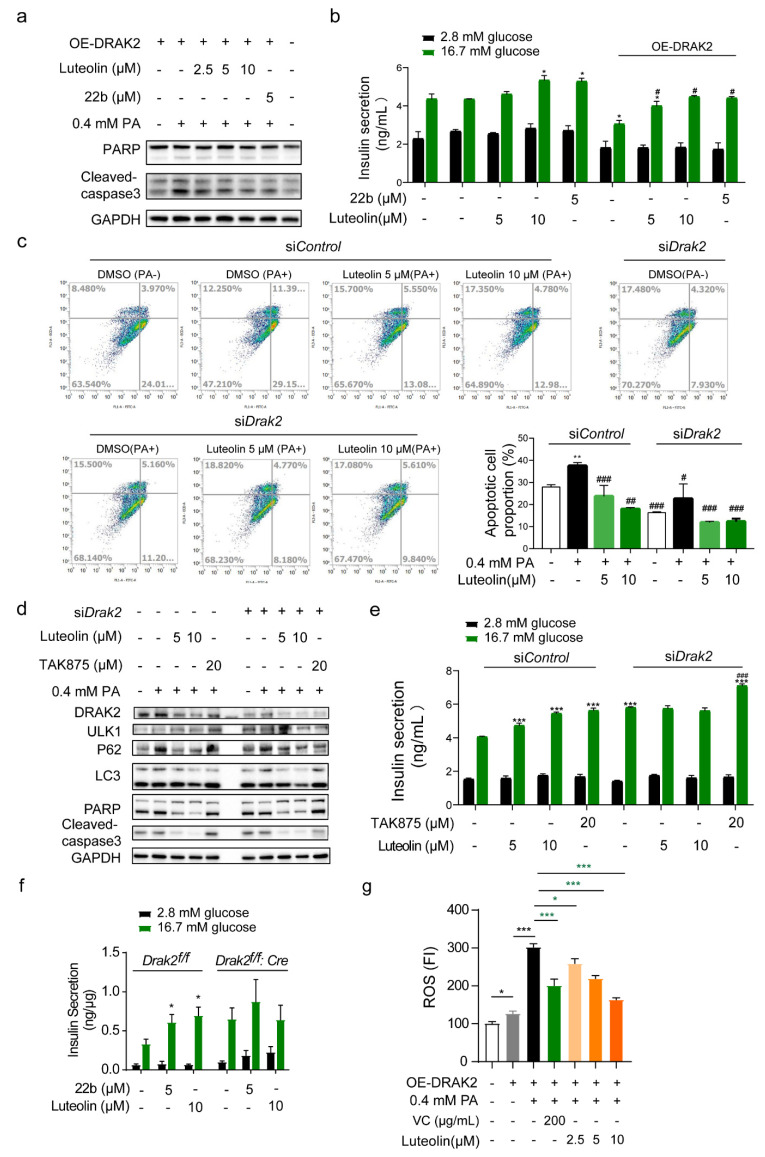
Effects of luteolin on apoptosis in INS-1E cells and GSIS function in primary islets. (**a**) Apoptosis marker proteins PARP and Cleaved-caspase 3 levels in INS-1E. (**b**) GSIS analysis on INS-1E cells after *Drak2* overexpression, n = 3, * *p* < 0.05, vs DMSO group. # *p* < 0.05, vs OE-*Drak2* DMSO group. (**c**) Representative FACS analysis and quantification of apoptotic rate after si*Drak2* in INS-1E cells, n = 3. ** *p* < 0.001, vs si*Control* DMSO group, # *p* < 0.05, ## *p* < 0.01, ### *p* < 0.001, vs. si*Control* PA group. (**d**) Western blot analysis of PRAP and Cleaved-caspase 3 protein levels in INS-1E cells after si*Drak2*. (**e**) GSIS analysis in INS-1E cells after si*Drak2*, n = 3. *** *p* < 0.001, vs DMSO group, ### *p* < 0.05, vs si*Drak2* DMSO group. (**f**) GSIS analysis on primary islet, which isolated from *Drak2^f/f^* mice and *Drak2^f/f^: Cre* mice, were treated by luteolin and 22b for 90 min, n = 3, * *p* < 0.05, vs *Drak2^f/f^* mice DMSO group. (**g**) ROS levels after overexpressing DRAK2 and treatment of Vitamin C (VC) or luteolin in PA-induced INS-1E cells for 24 h, n = 5. All values are presented as mean ± SEM of at least two independent experiments * *p* < 0.05, *** *p* < 0.001.

**Figure 4 pharmaceuticals-16-00975-f004:**
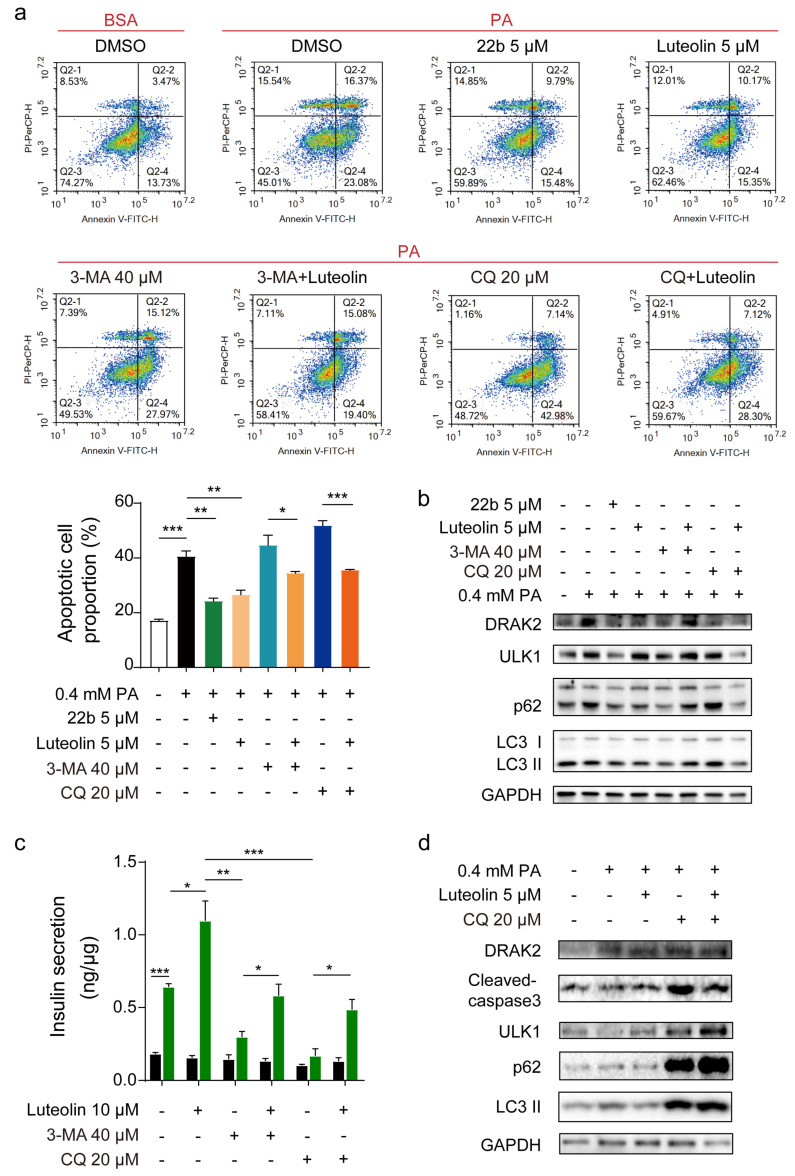
Therapeutic effects of luteolin reduced by autophagy inhibitors. (**a**) Representative FACS analysis and quantification of apoptotic rate with PA induced in response to various treatments in INS-1E cells. CQ and 3-MA are inhibitors of autophagy and thus serve as negative controls; 22b is an inhibitor of *Drak2* and thus serves as a positive control, n = 3. (**b**) Western blot analysis of the autophagy marker proteins ULK1, LC3 II and p62 levels in INS-1E cells. (**c**) GSIS on primary islets after treatment, measured by HTRF assay, n = 3. (**d**) Western blot analysis of apoptosis marker protein on primary islets. All values are presented as mean ± SEM of at least three independent experiments. * *p* < 0.05, ** *p* < 0.01, *** *p* < 0.001.

**Figure 5 pharmaceuticals-16-00975-f005:**
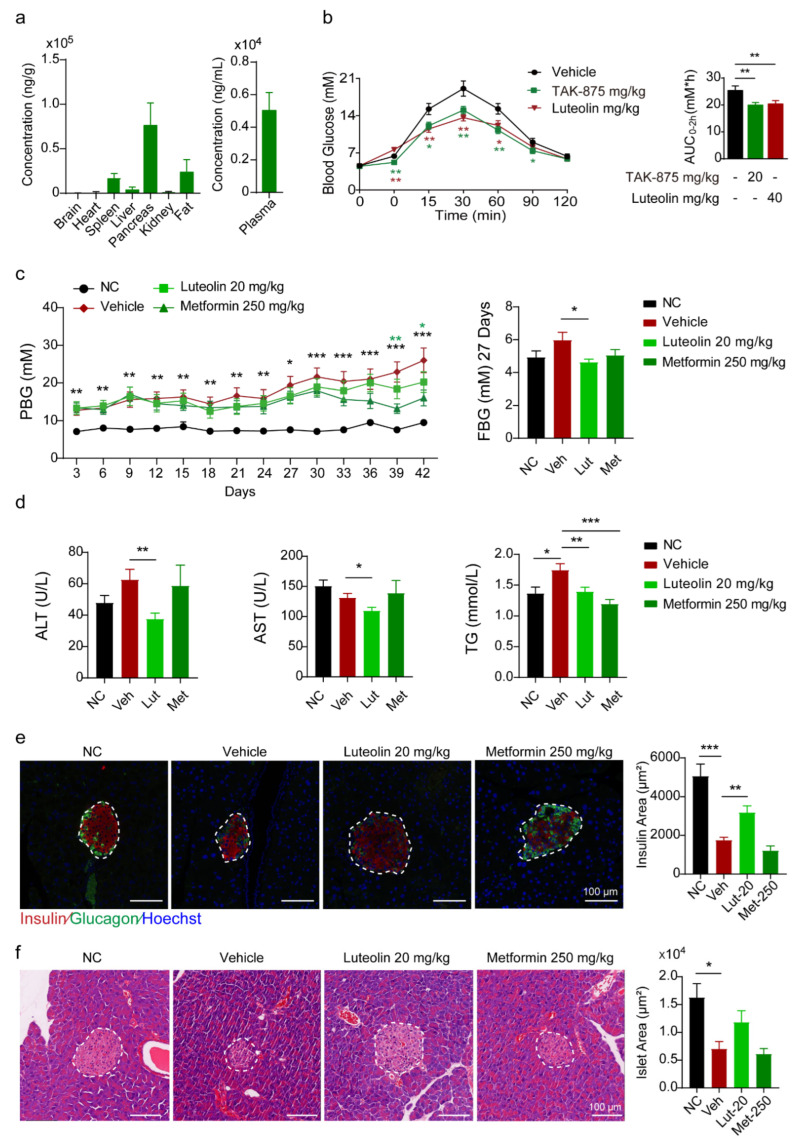
Therapeutic effects of luteolin on mice. (**a**) Tissue distribution of luteolin by i.p. in ICR mice, n = 4. (**b**) GTT analysis and quantification on vehicle and compound treatment groups. Blood glucose was determined at the indicated time points after i.p. injection of 2 g/kg glucose (n = 10 per group) and AUC0-120min. (**c**) Postprandial blood glucose (PBG) levels in different treatment groups, and fasting blood glucose (FBG) level at 27 days in different treatment groups, n = 8–13. (**d**) Differences in ALT, AST, TG in plasma between different treatment groups, n = 8–13. (**e**) Immunofluorescence of glucagon (green) and insulin (red) in pancreatic islets of mice, n = 20. (**f**) Area of islet assessed by hematoxylin and eosin (H&E) staining, n = 20. All values are presented as mean ± SEM of at least three independent experiments; * *p* < 0.05, ** *p* < 0.01, *** *p* < 0.001.

## Data Availability

Data is contained within the article and Appendix A.

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
