# Peer review of "Luteolin Protects Pancreatic β Cells against Apoptosis through Regulation of Autophagy and ROS Clearance"

_pharmaceuticals, 2023, doi:10.3390/ph16070975_

Round 1

Reviewer 1 Report

The Introduction, the aim of the work as well as the Discussion part must be rewritten in a more comprehensive way.

The Figures are too small and hard to read and need to be more clearly explained.

Although a lot of experiment have been done in the experimental part, the results are not clearly presented.

The text contains many syntactic ana grammatical errors and must be rewritten and resubmitted.

Author Response

We sincerely thank you for your valuable feedback that that help us to improve the quality of our manuscript. The comments are laid out below in italicized font and specific concerns have been numbered. Our response is given in normal font.

  1. The Introduction, the aim of the work as well as the Discussion part must be rewritten in a more comprehensive way.

Response: Thanks a lot for your suggestions. We have rewritten each part of our manuscript and added more detailed explanations of the aim to our work.

  1. The Figures are too small and hard to read and need to be more clearly explained.

Response: Thanks for your suggestion. We have reset the size of the figures and interpreted the explanations more thoroughly.

  1. the results are not clearly presented.

Response: We have reanalyzed the data more carefully and reorganized our results to present our purport as clearly as possible.

  1. The text contains many syntactic ana grammatical errors and must be rewritten and resubmitted.

Response: Thank you for the detailed review. We have carefully and thoroughly proofread the manuscript to correct the grammar and typos.

Reviewer 2 Report

Abstract- avoid abbreviations. Line 15- if possible, write the whole name instead of DAP.

Also, when using the keyword, include the whole name for keyword if possible.

Introduction- highlight all aspects of the research topic and use appropriate references. If possible, include newest onoes. The introduction gives a clear insight into the research topic of the paper, it is clearly structured and presents the topic concisely.

Results and Discussion

The results of the paper are clearly presented and described, and any images and graphical representations used are necessary for clearer monitoring of the results. In the discussion, there is no unnecessary repetition of results explanation, and the authors have clearly explained the results obtained and critically compared them with some other relevant research. In this context, I only suggest that the discussion part of the paper should be expanded in terms of a stronger link between the positive effects of luteolin on ROS, i.e., look for more relevant research that proved this.

Materials and Methods

They are clearly described and provide sufficient information for reproducibility of the experiments. Statistical methods are used appropriately.

The conclusion of the paper is omitted. Given the novelty of the results, I suggest that the authors include a Conclusion section that reiterates the major accomplishments of this work.

Author Response

We sincerely thank you for your valuable feedback that that help us to improve the quality of our manuscript. The comments are laid out below in italicized font and specific concerns have been numbered. Our response is given in normal font.

  1. avoid abbreviations. Line 15- if possible, write the whole name instead of DAP. Also, when using the keyword, include the whole name for keyword if possible.

Response: Thanks for your kindly advices. We have revised our manuscript to make sure all abbreviations were explained when first mentioned.

  1. highlight all aspects of the research topic and use appropriate references. If possible, include newest ones. The introduction gives a clear insight into the research topic of the paper, it is clearly structured and presents the topic concisely.

Response: We fully agree with your precious suggestions and the references have been updated accordingly.

  1. In this context, I only suggest that the discussion part of the paper should be expanded in terms of a stronger link between the positive effects of luteolin on ROS, i.e., look for more relevant research that proved this.

Response: Thank you for your thoughtful advice. We have added more contents to explain the inner link between luteolin and its antioxidant effect in discussion part.

  1. The conclusion of the paper is omitted. Given the novelty of the results, I suggest that the authors include a Conclusion section that reiterates the major accomplishments of this work.

Response: Thanks for your comments and suggestions. We summarize the innovations and results of the manuscript in the conclusion.

Reviewer 3 Report

The authors presented a paper entitled Luteolin Protects Pancreatic β Cell Against Apoptosis Through Autophagy Process Regulation and ROS Clearance.

The topic is interesting and within the aims and scopes of the Journal.

Yet, the manuscript needs several major implementations and changes (especially in the writing, style and discussion) before it can be really considered for publication in this Journal.

My comments are reported below one by one:

- All the paper: Please use the past tenses for every verb when you describe your results and please avoid the term we. Modify all the sentences into passive or impersonal. In addition, some verbs are not in the correct tense. Things that are always valid must have the verb in present tense.

- The abstract must be completely revised. It is poorly written.

- Lines 35-37: What does this mean exactly?

- Lines 41-44: What?

- Line 49: PA?

- The introduction section is fine, but it must be more linearly presented.

- What about the existing synthetic compounds for treating this problem and their eventual side effects and problems in recovery, pharmacodynamics and production? This aspect must be developed in the Introduction Section.

- Line 69: A single compound is not a medicine if not part of a complex formulation. This part must be checked.

- Lines 69-72: Nonsense as written. In addition, what are the general efficacy values of this compound for each activity? Please report them.

- Is this research new or not? This aspect must be cleared.

- What is exactly the background for this research? Why did you choose to study this aspect? All these things must be better explained.

- Lines 87-88: You must at least cite all these compounds you individuated for this thing. In addition, you must better justify the reason why you focused on luteolin. Was it only a matter of values? According to this, what were the values for all the other compounds? A range for this must be given, at least.

- Lines 89-92: What?

-  It is not clear where you recovered luteolin. I really hope you recovered it from a natural source. If so, you must specify in the first lines of the Section 2, the methodology adopted for its extraction, isolation and purification with the relative values. In contrast, I really hope you did not buy it. I remind you that synthetic and natural compounds may have different effects in vitro and in vivo. Nevertheless, I really hope this work is not only theoretical.

 - Line 101: 22b?

 - You must present the values for the percentage and the IC50 values for all the things reported in Sections 2.2, 2.4, 2.5.

- Lines 122: Nonsense.    

- Lines 125-128: What?

- Line 140: In vitro must be in Italics.

- Lines 178-179: What?

- Lines 206-207: What?

- Lines 227-229: What?

- Line 232: What does this exactly mean?

- Lines 239-245: What?

- Lines 246-252: This is not for Section 3 but rather for Section 1.

- “Chinese traditional medicine has unique advantages...” In what sense and what are these advantages?

- Lines 250-252: What?

- Lines 257-263: What?

- Lines 265-266: And why did you not study these aspects, too?

- What about comparing the values and the mechanisms of action with other similar compounds?

- Line 272: Where did you take all these compounds?

- Section 4 must be written with real sentences and not as hints. The same style must be adopted as well as the same verb tenses.

- Line 338: Citation missing here.

- All the manuscript: Several sentences are too long and/or poorly written. Please separate the relative results from the introductive sentences.

- All the references are not exactly cited in the text as requested by the Journal.

- All the references are not exactly written in the list as requested by the Journal.

Author Response

We sincerely thank you for your valuable feedback that that help us to improve the quality of our manuscript. The comments are laid out below in italicized font and specific concerns have been numbered. Our response is given in normal font.

  1. All the paper: Please use the past tenses for every verb when you describe your results and please avoid the term we. Modify all the sentences into passive or impersonal. In addition, some verbs are not in the correct tense. Things that are always valid must have the verb in present tense.

Response: Thanks for your comments and suggestions. These mistakes have been corrected in revised manuscript.

  1. The abstract must be completely revised. It is poorly written.

Response: Thank you for pointing out our deficiencies and we are sorry for our poor writings. A native English speaker from US has been invited to polish our manuscript and the context has been reorganized in a more comprehensive way. We hope the revised version could be acceptable for you.

  1. Lines 35-37: What does this mean exactly?

Response: In this part, we proposed that β cell apoptosis accounted for the progression of diabetes mostly, but few of the existing anti-diabetic drugs could solve this problem except for GLP-1 related drugs. Therefore, it is necessary to further develop hypoglycemic compounds which have the efficacy of alleviating β cells apoptosis. (Adjusted to lines 39-42)

  1. Lines 41-44: What?

Response: This part aimed to give an outlined description of the biological process of autophagy. We apologize for the inaccuracy statements and we have clarified this in the newest version. (Adjusted to lines 89-92)

  1. Line 49: PA?

Response: Thanks for your question. PA means palmitate (line 21). We apologize for not writing the full name when the abbreviation was first mentioned. We have checked full text to make sure all abbreviations were explained at the first mention.

  1. The introduction section is fine, but it must be more linearly presented.

Response: Thanks for your suggestion. We have reorganized and revised the introduction section.

  1. What about the existing synthetic compounds for treating this problem and their eventual side effects and problems in recovery, pharmacodynamics and production? This aspect must be developed in the Introduction Section.

Response: Thanks a lot for your precious suggestion. We have added to the introduction the deficiencies of these compounds for diabetes. (Details in lines 43-45)

  1. - Line 69: A single compound is not a medicine if not part of a complex formulation. This part must be checked.

Response: Thank you very much for pointing out this mistake. We have modified this phrase in the manuscript as follows:

“Luteolin, a polyphenolic bioflavonoid, exists in multiple kinds of Chinese traditional medicine. It has been reported to have……in cells.” (Details in lines 336-344)

  1. - Lines 69-72: Nonsense as written. In addition, what are the general efficacy values of this compound for each activity? Please report them.

Response: Thanks a lot for your advice. We added general efficacy values for the corresponding effects according to references. “anti-tumor” in vitro was 20 µM. “Anti-inflammation” in cell was used as 100 mM, and in animal was 20/50 mg/kg. “Anti-oxidant” use was 5/10 mM in cells and 20 mg/kg in animal. “Anti-allergy” in animal was 50 mg/kg and “anti-bacterial” was used 5-50 μM in vitro and 10-100 mg/kg in vivo. “Anti-diabetic” use was 100 mg/kg in mouse. In autophagy regulation use were 5/00 mM in cells, respectively. (Details in lines 338-344)

  1. - Is this research new or not? This aspect must be cleared.

Response: Thank you for reminding, this research is new and we have clarified this in our revised version.

  1. - What is exactly the background for this research? Why did you choose to study this aspect? All these things must be better explained.

Response: Thank you for your advice. As you kindly suggested, we clarified the specific background of our study as well as the reason for choosing such topic. The introduction part has been modified.

  1. - Lines 87-88: You must at least cite all these compounds you individuated for this thing. In addition, you must better justify the reason why you focused on luteolin. Was it only a matter of values? According to this, what were the values for all the other compounds? A range for this must be given, at least.

Response: Thanks for your suggestion. Aiming to discover the natural product has the potential DRAK2 inhibitory activity, we screened the natural products focus library provided by Professor Jianmin Yue from Shanghai Institute of Materia Medica, Chinese Academy of Sciences. From the primary screening with concentration of 20 μM, we found the Y03 (Luteolin) with the best DRAK2 inhibitory activity, but the others did not affect much, as information below shows.

Based on background survey, the hypoglycemic efficacy of luteolin has been reported in other studies before. (Zang Y, etc., Biosci Biotechnol Biochem. 2016 Aug;80(8):1580-6.), then we picked the luteolin for the in vitro functional assay and in vivo efficacy study.

Sample ID

Concentration

Result Type

Result

Y03

20 μM

%Activity

7.74  +/- 1.75

RLB-1

20 μM

%Activity

94.17 +/- 6.84

RLB-2

20 μM

%Activity

66.57 +/- 5.02

RLB-3

20 μM

%Activity

90.62 +/- 14.87

RLB-5

20 μM

%Activity

54.19 +/- 1.74

RLB-6

20 μM

%Activity

66.35 +/- 6.03

RLB-8

20 μM

%Activity

68.73 +/- 7.78

RLB-9

20 μM

%Activity

79.49 +/- 4.96

RLB-10

20 μM

%Activity

64.85 +/- 1.14

RLB-11

20 μM

%Activity

71.04 +/- 6.00

RLA-4

20 μM

%Activity

89.42 +/- 5.85

RLO-4

20 μM

%Activity

37.86 +/- 6.00

RLO-5

20 μM

%Activity

57.94 +/- 4.99

RLO-6

20 μM

%Activity

62.09 +/- 4.02

RLO-7

20 μM

%Activity

82.72 +/- 3.50

RLO-9

20 μM

%Activity

58.23 +/- 0.59

RLO-10

20 μM

%Activity

50.13 +/- 8.84

RLO-11

20 μM

%Activity

42.79 +/- 1.24

RLO-12

20 μM

%Activity

67.75 +/- 2.59

RLO-13

20 μM

%Activity

39.83 +/- 3.97

RLO-14

20 μM

%Activity

41.67 +/- 0.74

RLO-15

20 μM

%Activity

46.50 +/- 0.31

RLO-16

20 μM

%Activity

70.31 +/- 0.46

RLO-17

20 μM

%Activity

56.54 +/- 1.82

RLO-18

20 μM

%Activity

64.38 +/- 10.76

RLO-19

20 μM

%Activity

62.46 +/- 10.21

RLO-20

20 μM

%Activity

75.10 +/- 7.60

RLM- 1

20 μM

%Activity

35.54 +/- 5.27

HZ-C5

20 μM

%Activity

132.98 +/- 8.06

HZ-C3

20 μM

%Activity

99.30 +/- 17.26

HZ-C31

20 μM

%Activity

114.00 +/- 3.33

HZ-C32

20 μM

%Activity

100.80 +/- 3.01

HZ-C6

20 μM

%Activity

100.85 +/- 3.11

HZ-C7

20 μM

%Activity

94.30 +/- 8.89

HZ-C10

20 μM

%Activity

96.97 +/- 12.62

HZ-C14

20 μM

%Activity

87.74 +/- 4.46

HZ-C29

20 μM

%Activity

87.24 +/- 3.42

HZ-C1

20 μM

%Activity

101.29 +/- 9.46

HZ-C19

20 μM

%Activity

121.20 +/- 4.76

HZ-C20

20 μM

%Activity

107.05 +/- 14.73

HZ-C23

20 μM

%Activity

124.97 +/- 1.60

HZ-C24

20 μM

%Activity

117.22 +/- 1.77

HZ-C50

20 μM

%Activity

120.93 +/- 1.52

HZ-C9

20 μM

%Activity

90.94 +/- 5.62

HZ-C13

20 μM

%Activity

115.96 +/- 2.70

HZ-C12

20 μM

%Activity

91.04 +/- 10.28

HZ-C16

20 μM

%Activity

84.23 +/- 5.15

HZ-C11

20 μM

%Activity

77.85 +/- 8.63

HZ-C8

20 μM

%Activity

111.22 +/- 17.70

HZ-C15

20 μM

%Activity

109.61 +/- 10.77

HZ-C30

20 μM

%Activity

124.59 +/- 2.44

HZ-C33

20 μM

%Activity

76.80 +/- 2.28

HZ-C40

20 μM

%Activity

103.10 +/- 8.59

HZ-C17

20 μM

%Activity

84.57 +/- 5.67

HZ-C2

20 μM

%Activity

101.27 +/- 13.42

HZ-C42

20 μM

%Activity

106.75 +/- 12.79

HZ-C34

20 μM

%Activity

95.42 +/- 0.30

HZ-C48

20 μM

%Activity

81.72 +/- 3.59

HZ-C43

20 μM

%Activity

94.56 +/- 1.68

HZ-C44

20 μM

%Activity

71.32 +/- 4.53

HZ-C45

20 μM

%Activity

72.82 +/- 8.92

HZ-C46

20 μM

%Activity

59.71 +/- 16.52

HZ-C47

20 μM

%Activity

97.67 +/- 6.70

HZ-C35

20 μM

%Activity

63.98 +/-11.92

HZ-C36

20 μM

%Activity

70.29 +/- 4.25

HZ-C37

20 μM

%Activity

50.18 +/- 7.75

HZ-C38

20 μM

%Activity

63.99 +/- 5.36

HZ-C39

20 μM

%Activity

45.53 +/- 6.25

HZ-C57

20 μM

%Activity

109.61 +/- 0.87

HZ-C58

20 μM

%Activity

48.07 +/- 7.82

HZ-C59

20 μM

%Activity

55.53 +/- 5.60

HZ-C60

20 μM

%Activity

55.53 +/- 1.31

HZ-C61

20 μM

%Activity

56.78 +/- 4.34

HZ-C18

20 μM

%Activity

100.32 +/- 17.82

HZ-C41

20 μM

%Activity

75.63 +/- 5.23

HZ-C49

20 μM

%Activity

65.46 +/- 0.09

HZ-C52

20 μM

%Activity

74.12 +/- 11.60

HZ-C53

20 μM

%Activity

93.81 +/- 2.81

RL4O-3

20 μM

%Activity

71.46 +/- 16.58

RL4O-5

20 μM

%Activity

56.37 +/- 0.89

RL4O-11

20 μM

%Activity

109.71 +/- 0.94

RL4O-13

20 μM

%Activity

58.04 +/- 0.67

RL4O-14

20 μM

%Activity

82.03 +/- 6.81

RL4O-16

20 μM

%Activity

72.54 +/- 14.05

RL4O-18

20 μM

%Activity

66.28 +/- 0.28

RL4O-19

20 μM

%Activity

88.22 +/- 8.15

RL4O-20

20 μM

%Activity

49.55 +/- 1.71

RL4O-21

20 μM

%Activity

138.25 +/- 3.21

RL4O-22

20 μM

%Activity

63.95 +/- 14.49

RL4O-23

20 μM

%Activity

117.94 +/- 6.12

RL4O-24

20 μM

%Activity

104.99 +/- 13.44

RL4O-25

20 μM

%Activity

85.20 +/- 32.13

RL4O-26

20 μM

%Activity

82.90 +/- 16.23

RL4O-27

20 μM

%Activity

52.99 +/- 16.55

RL4O-28

20 μM

%Activity

83.17 +/- 2.80

RL4O-30

20 μM

%Activity

88.82 +/- 1.53

RL4O-32

20 μM

%Activity

52.73 +/- 0.34

RL4O-33

20 μM

%Activity

51.93 +/- 1.04

H-67

20 μM

%Activity

101.06 +/- 3.01

H-78

20 μM

%Activity

81.88 +/- 3.97

H-92

20 μM

%Activity

96.82 +/- 0.38

HES-105

20 μM

%Activity

75.79 +/- 1.07

HES-106

20 μM

%Activity

95.94 +/- 7.65

HES-78

20 μM

%Activity

64.84 +/- 14.20

APO-10b2

20 μM

%Activity

76.28 +/- 6.44

APO-9

20 μM

%Activity

69.88 +/- 2.65

APO-5

20 μM

%Activity

59.83 +/- 4.16

JAZ-3-1

20 μM

%Activity

75.20 +/- 2.08

JAZ-18-4-2

20 μM

%Activity

77.45 +/- 4.17

JAZ-4-B1

20 μM

%Activity

65.93 +/- 1.86

JAZ-7-1-2

20 μM

%Activity

80.21 +/- 5.84

JAZ-3-2-2

20 μM

%Activity

83.28 +/-6.67

JAZ-6-5A

20 μM

%Activity

77.86 +/- 3.77

JAZ-6-5B

20 μM

%Activity

78.61 +/- 8.16

JAZ-2-1A

20 μM

%Activity

77.70 +/- 9.38

JAZ-2-1B

20 μM

%Activity

70.58 +/- 9.38

JAZ-31-5

20 μM

%Activity

63.39 +/- 6.90

JAZ-31-3

20 μM

%Activity

50.37 +/- 6.94

The primary screening protocol and detailed methods were described in 5.1.

  1. - Lines 89-92: What?

Response: Thank you for pointing out our misleading expression. We aimed to describe the molecular inhibition activity of luteolin on DRAK2, and we used 22b as our positive control to compare the efficacy of luteolin on DRAK2 inhibition. (Details in lines 123-127)

  1. -  It is not clear where you recovered luteolin. I really hope you recovered it from a natural source. If so, you must specify in the first lines of the Section 2, the methodology adopted for its extraction, isolation and purification with the relative values. In contrast, I really hope you did not buy it. I remind you that synthetic and natural compounds may have different effects in vitro and in vivo. Nevertheless, I really hope this work is not only theoretical.

Response: Thanks for your comment to emphasize the distinct characterization of natural product from extraction or totally synthesized. I’m completely agree with your opinion, and the activity (especially in vivo efficacy activity) will be greatly affected by the methodology of extraction, isolation and purification. To avoid this, we confirmed firstly the inhibitory activity on DRAK2 with the two batches of luteolin, and no significant difference was observed.

For the cellular and animal experiments, luteolin (HY-N0162) was purchased in powder form. For original screening, luteolin was provided by Professor Jianmin Yue (Shanghai Institute of Materia Medica, Chinese Academy of Sciences). The structural information has been displayed in Figure 1a.

  1. - Line 101: 22b?

Response: 22b is a DRAK2 inhibitor which was reported before. We used it as the positive control and we have mentioned it in 2.1 (line 127) and have cited the corresponding reference accordingly.

  1. - You must present the values for the percentage and the IC50 values for all the things reported in Sections 2.2, 2.4, 2.5.

Response: Thanks for reminding. We have modified this in our manuscript. (Details in lines 138-141, lines 166-170 and lines 219-222)

  1. - Lines 122: Nonsense.

Response: As mentioned, overexpression DRAK2 and knockdown Drak2 INS-1E cells were used as positive and negative controls, respectively. In this way, we would like to verify if DRAK2 took part in PA-induced β cell damage and whether luteolin could protect β cell from such damage partially through inhibiting DRAK2. (Details in lines 163-164)

  1. - Lines 125-128: What?

Response: We aimed to clarify whether the protection efficacy of luteolin on INS-1E cells partially dependent on DRAK2 inhibition. We used flow cytometry assay (FACS) to harvest the Drak2-knockdown cells and performed following experiments. It was found that Drak2 knockdown had weakened the effect of luteolin, which proved our assumption. Thank you for your reminding and we have revised the description in our manuscript. (Details in lines 170-174)

  1. - Line 140: In vitro must be in Italics.

Response: Thank you for your thoughtful reminding. We have modified in our manuscript. (Details in line 187)

  1. - Lines 178-179: What?

Response: Chloroquine (CQ), a lysosomotropic agent, is commonly used to inhibit lysosomal degradation and autophagy. When adding to cells, it causes accumulation of lysosomal marker protein such as p62, LC3 II, etc. As our result illustrated, adding CQ in PA-treatment cells would inhibit the autophagic flux and autophagy marker proteins were increased. However, marker proteins further accumulated after luteolin treatment, which meant the autophagic flux was promoted by luteolin. Thanks for your inquiry and we have revised our description. (Details in lines 226-230)

  1. - Lines 206-207: What?

Response: Metformin, as an effective and fundamental hypoglycemic agent, is now widely used in clinical treatments and researches for diabetes. Therefore, we chose metformin as our positive control in comparison of the hypoglycemic capacity of luteolin. In our study, metformin effectively reduced the blood glucose of diabetic mice, which was the same result we expected and as other researches had suggested. So, we mentioned this result in this section. Thanks for pointing out that metformin should be explained when firstly mentioned. Now we have clarified this in our manuscript. (Details in lines 256-257)

  1. - Lines 227-229: What?

Response: Natural plants in traditional Chinese medicine have plenty of medical products, and luteolin may be one of them. So, we would like to stress the potential of natural products as well as luteolin and explain the reason why we screened natural products library for DRAK2 inhibitor. And we have adjusted the description. (Details in lines 332-335)

  1. - Line 232: What does this exactly mean?

Response: Luteolin is a polyphenolic bioflavonoid. Its structure contains multiple phenolic hydroxyls, which implies it may exhibit reducibility. Since most other flavonoids has antioxidant function, we speculated that luteolin may also be antioxidative. The sentence has been adjusted and we have added explanation. (Details in lines 362-363)

  1. - Lines 239-245: What?

Response: DRAK2 belongs to the DAPK family. It is reported in other researches that many DAPK family member proteins are involved in autophagy processes, which means DRAK2 potentially participate, yet haven’t been studied. Luteolin is an inhibitor of DRAK2. According to our result, it may promote β cell autophagy flux partially through DRAK2 inhibition, which means DRAK2 may directly involve in the regulation of autophagy process. Further study is still needed. We hope our research would be inspirations for follow-up research and we will also conduct deeper researches on this aspect. We discussed our speculation more detail in the revision, as displayed in paragraph 6 of Discussion part. (Details in lines 379-396)

  1. - Lines 246-252: This is not for Section 3 but rather for Section 1.

Response: Thanks for your thoughtful advice. This has been moved to Section 1. (Details in lines 70-88)

  1. - “Chinese traditional medicine has unique advantages...” In what sense and what are these advantages?

Response: We have deleted this phrase and reorganized our manuscript. Thank you for your precious suggestion. (Details in lines 79-88)

  1. - Lines 250-252: What?

Response: We have revised this part and thank you for your precious suggestion. In this part, we stressed the multiple therapeutic effects of natural products instead of Chinese medicine. (Details in lines 80-87)

  1. - Lines 257-263: What?

Response: Thanks for inquiry. We have reorganized the logic of our manuscript. We adjusted this position in the discussion. (Details in lines 312-320)

  1. - Lines 265-266: And why did you not study these aspects, too?

Response: We must admit there still exists plenty of work to explain the detailed mechanism of luteolin, but in matter of fact, this would take a long time, so it is not currently discussed in the manuscript. As mentioned in our manuscript, orally treatment of luteolin had poor bioavailability and eliminated in short time. We are now working with our collaborators to have luteolin structurally modified, aiming for obtaining more active compounds. These aspects would be further discussed in our future researches. (Details in lines 400-402)

  1. - What about comparing the values and the mechanisms of action with other similar compounds?

Response: Luteolin was one of the most effective DRAK2 inhibitors among those natural products we had screened. This is the novelty of our study and the reason why we chose luteolin for deeper research. We did not mention other compounds for there are still few researches of DRAK2 relating to those kinds of compounds. Thank you for your thoughtful suggestion. Now we have added some discussions of other similar compounds.

  1. - Line 272: Where did you take all these compounds?

Response: The natural products focus library provided by Professor Jianmin Yue (Shanghai Institute of Materia Medica, Chinese Academy of Sciences).

  1. - Section 4 must be written with real sentences and not as hints. The same style must be adopted as well as the same verb tenses.

Response: Thanks for your precious suggestion. We have revised this in our manuscript.

  1. - Line 338: Citation missing here.

Response: Thanks for your reminding. The corresponding references have been added at 5.7 (line 472) and the literature citations were checked as well.

  1. - All the manuscript: Several sentences are too long and/or poorly written. Please separate the relative results from the introductive sentences.

Response: Thanks for your suggestion. We have revised the manuscript to make sentences more accurate and easier to understand.

  1. - All the references are not exactly cited in the text as requested by the Journal.

Response: Thank you for carefully reminding. We have revised our citation format.

  1. - All the references are not exactly written in the list as requested by the Journal.

Response: Thank you for reminding. The citation list has been revised accordingly.

Reviewer 4 Report

The English in this work is severely flawed, which prevents its complete evaluation. I was unable to understand most of the introduction and discussion. Unless the others markedly improve it, this work will be recommended for rejection on the grounds of unintelligibility.

Some comments can be provided already:

The authors mention natural products “multi-efficacy”. This concept is vague and must be clarified.

The abstract is a collection of sentences that are not correctly conjugated and requires significant revision.

Densitometric analysis of western blot results is mandatory.

Figure 4d and remaining WBs requires higher resolution to be properly assessed.

% of vechicles must be clearly stated in all cases.

 Scale in 5f is not clear and needs revision.

Please make the submission more readable for subsequent revision.

Author Response

We sincerely thank you for your valuable feedback that that help us to improve the quality of our manuscript. The comments are laid out below in italicized font and specific concerns have been numbered. Our response is given in normal font.

  1. The authors mention natural products “multi-efficacy”. This concept is vague and must be clarified.

Response: Thanks for your suggestion. “multi-efficacy” in manuscript is means that the characteristics of natural products in preventing and treating diabetes are multi-link, multi-pathway and multi-target. For example, different from Rosiglitazone (RSG) which only reduces blood glucose, extracts of Chinese herbs such as Coptis chinensis and Salvia miltiorrhiza strengthen insulin sensitivity, decrease visceral fat, improve hyperlipidemia, and regulate body functions simultaneously.

  1. The abstract is a collection of sentences that are not correctly conjugated and requires significant revision.

Response: Thanks a lot for your precious suggestions. we have revised the details in the abstract section and rearranged statements according to logic.

  1. Densitometric analysis of western blot results is mandatory.

Response: Thanks for your suggestion. We added the densitometric analysis statistics of western blot results in the supplementary material of the corresponding Figures.

  1. Figure 4d and remaining WBs requires higher resolution to be properly assessed.

Response: Thank you for the suggestion. The western blot sample in Figure 4d was harvest from mouse primary islets. The proteins of primary islets were so complex that we were not able to obtain higher quality results. But we have done a densitometric analysis of the results multiple times, and we hope the revised manuscript could be acceptable for you.

  1. % of vehicles must be clearly stated in all cases.

Response: Thanks a lot for your precious suggestions. We added the percentage of vehicles to the description of the results in all cases.

  1. Scale in 5f is not clear and needs revision.

Response: Thanks for your precious comments. We modified the scale in Figure 5f to make it clearer.

Round 2

Reviewer 1 Report

The text and the phrases added by the authors to the text have to be written in a more clear way and checked for grammatical / syntactical errors, especially page 2 (lines 79-88), page 3 (lines 123-129), page 4 (lines 138-141), page 7 (lines 166-170) as well as the Conclusion part.

Author Response

Thank you very much for your advice. We have revised the manuscript, and would like to re-submit it for your consideration. The comments are laid out below and highlighted in red. Our responses are presented in normal fonts.

  1. The text and the phrases added by the authors to the text have to be written in a more clear way and checked for grammatical/syntactical errors, especially page 2 (lines 79-88), page 3 (lines 123-129), page 4 (lines 138-141), page 7 (lines 166-170) as well as the Conclusion part.

Response: Thank you for pointing out and we have modified these parts in the newest version. We invited the American journal expert to help us improving the language. Each part is now presented as follows:

Page 2 (lines 79-88 now in lines 89-98)

“Flavonoids are widely distributed phytochemicals in dietary plants and Chinese herbal medicines. Many studies have found that most flavonoids exhibit excellent antioxidant activity, and some dietary flavonoids show promising antidiabetic and hypoglycemic effects as well [23]. This suggests that natural products may be of great benefit for the protection of pancreatic islet β cells. Luteolin, as a natural product belonging to the group of dietary flavonoids, also has antioxidant effects [15]. A previous study demonstrated that luteolin could reduce the formation of nitric oxide (NO) and inducible nitric oxide synthase (iNOS). It also upregulated the expression of the transcription factor MafA in β cells, thus increasing the secretion of insulin in urea-damaged β cells [24], which proved that luteolin may also protect islet β cells via its antioxidant activity.”

Page 3 (lines 123-129 now in lines 132-143)

“The natural products we screened were listed in the Supplementary Table S1. ADP-GloTM Kinase Assay in vitro showed that part of these compounds affected DRAK2 activity with more than 50% inhibitory rate at the primary screened concentration of 20 μM. Among them, luteolin showed the strongest effect with an IC50 of 346.7 +/- 30.0 nM (Figure 1a, b). Compound 22b was reported to be an effective DRAK2 inhibitor, so it was used as the positive control in this study [25]. As mentioned above, luteolin also has been reported with antidiabetic effects. Therefore, we assumed it was worth further investigating the mechanism underlying the association between its inhibitory effect on DRAK2 and islet β cell protection.”

Page 4 (lines 138-141 now in lines 151-155)

“The cell apoptosis rate increased markedly to 55.79% in INS-1E cells after PA treatment, while the apoptosis rate decreased to 36.71% after treatment of 22b (5 μM) (Figure 2a). As Figure 2a shows, luteolin alleviated cell apoptosis in a dose-dependent manner, and the apoptosis rate declined from 48.09% (2.5 μM) to 44.42% (5 μM) and then 35.83% (10 μM) (Figure 2a).”

Page 7 (lines 166-170 now in lines 185-195)

“The apoptosis rate increased from 27.98% (control group) to 37.51% after PA treatment, but 5 μM and 10 μM luteolin reduced it to 23.74% and 17.99%, respectively (Figure 3c). When DRAK2 was knocked down in INS-1E cells, the apoptosis rate was reduced to 12.25%, whereas it was 27.98% in the si-control group, and PA treatment only increased the apoptosis rate to 16.36%. This indicates that DRAK2 may participate in β cell apoptosis regulation. Nevertheless, after treatment with luteolin, the apoptosis rate was further lowered to 11.54% (5 μM) and 13.11% (10 μM), which suggests that luteolin can protect β cells from apoptosis, but this effect may not be entirely dependent on its inhibitory effect on DRAK2 (Figure 3c).”

Reviewer 3 Report

The authors presented a revised version of the paper I had previously reviewed.

The manuscript has improved since my last view but not enough to be accepted in its present form as shown in my comment below:

- Lines 15 and 17: Was? Is it no longer true now?

- Line 41: Failure?

- Line 82: The correct term is “They have”. The subject is flavonoids.

- Lines 84-86: Nonsense as written.

- Lines 87-88: Not all natural products. This sentence is too generic and misleading.

- Please report that PA is palmitic acid also in the Introduction section the first time you sue this acronym.

- Lines 123-124: Poorly written.

- Please write IC50 with 50 subscripted.

- Line 140: 22b again?

- The discussion section must be placed before the Conclusion Section.

- Many parts presented in Section 4 are not fine for this section. In addition, in this Section, you must write about your results and explain them also comparing your values with other similar compounds. All this part must be revised. There are too many introductive phrases.

- You must write v/v and in Italics.

- Your reply to my comment 12 must be presented in brief in the Results Section as a kind of Introduction to luteolin and the table must be presented as supplementary Material.

Author Response

Thank you very much for your advice. We have revised the manuscript, and would like to re-submit it for your consideration. The comments are laid out below and highlighted in red. Our responses are presented in normal fonts.

The authors presented a revised version of the paper I had previously reviewed. The manuscript has improved since my last view but not enough to be accepted in its present form as shown in my comment below:

Thank you very much for your advice. Regarding the concerns on language, we invited the American journal expert to help us edit the language of manuscript. Hope the revision will meet the criteria of publication.

  1. Lines 15 and 17: Was? Is it no longer true now?

Response: Thanks for pointing out this misleading and improper use. What we mentioned is an accepted fact and ‘is’ should be the right form other than ‘was’. This has been adjusted in the manuscript.  (Details on line 17)

  1. Line 41: Failure?

Response: Thank you for inquiry. ‘β cell failure’ here referred to β cell loss of function and mass. The same phrase can be referenced in other reports relating to this topic, e.g. (Talchai C, etc. Pancreatic β cell dedifferentiation as a mechanism of diabetic β cell failure. Cell. 2012 Sep 14;150(6):1223-34.; Eizirik DL, etc. Pancreatic β-cells in type 1 and type 2 diabetes mellitus: different pathways to failure. Nat Rev Endocrinol. 2020 Jul;16(7):349-362.). For better understanding, this word has been changed to ‘loss’ in the manuscript.  (Details in line 45)

  1. Line 82: The correct term is “They have”. The subject is flavonoids.

Response: Thanks a lot for your reminding. This section has been modified and mistakes as such have been corrected now.  (Details in lines 90-92)

  1. Lines 84-86: Nonsense as written.

Response: In this section, the known antioxidant function and underlying mechanisms of luteolin was introduced to lead to our following investigations on whether luteolin could protect β cell via reducing the oxidative stress. For better understanding, this section has been revised in the manuscript. (Details in lines 93-96)

  1. Lines 87-88: Not all natural products. This sentence is too generic and misleading.

Response: Thanks for your suggestion and the generic description has been adjusted in manuscript.  (Details in lines 90-93)

  1. Please report that PA is palmitic acid also in the Introduction section the first time you sue this acronym.

Response: The full name of PA in the introduction section has been added (Details in line 107). Thanks for your kindly reminding. Full-text abbreviations have been double-checked to avoid such mistake.

  1. Lines 123-124: Poorly written.

Response: Thanks for your suggestion and the manuscript has been revised. In this section, we described the inhibitory activity of luteolin against DRAK2. (Details in lines 129-132)

  1. Please write IC50 with 50 subscripted.

Response: Thank you for your reminding. We have modified this form in the manuscript.

  1. Line 140: 22b again?

Response: 22b is a known DRAK2 inhibitor as we mentioned in line-138. Previous study demonstrated that it could alleviate β cell apoptosis and improve GSIS, thus we chose it to compare the effect of luteolin and used it as a positive control in vitro.

  1. The discussion section must be placed before the Conclusion Section.

Response: Thanks for your kindly suggestion. Now the order has been adjusted.

  1. Many parts presented in Section 4 are not fine for this section. In addition, in this Section, you must write about your results and explain them also comparing your values with other similar compounds. All this part must be revised. There are too many introductive phrases.

Response: Thanks for the suggestion and this part has been modified. Introductions of mechanisms of flavonoids have been condensed and comparisons of the value of luteolin with other similar compounds have been added.

  1. You must write v/v and in Italics.

Response: Thank you for your reminding. We have modified the fonts in the manuscript.

  1. Your reply to my comment 12 must be presented in brief in the Results Section as a kind of Introduction to luteolin and the table must be presented as supplementary Material.

Response: Thanks for the suggestion. We have provided a brief description of the source of luteolin as well as the screening process of its molecular activity. The table contained all screened compounds is now available in supplemental materials.

Reviewer 4 Report

I had warned the authors before that the languague had to markedfy improved, which was not the case. "may also help anti-diabetes" in the abstract?!

In addition, I take note that the authors do not provide higher resolution images of their WB results, as requested. I was not asking for better bands (they are what they are), but better images of said results.

In light of this, I recommend this work to be rejected, on the grounds of lack of clarity of the language and part of the results.

Author Response

Thank you very much for your advice. We have revised the manuscript, and would like to re-submit it for your consideration. The comments are laid out below and highlighted in red. Our responses are presented in normal fonts.

  1. I had warned the authors before that the languague had to markedfy improved, which was not the case. "may also help anti-diabetes" in the abstract?!

Response: Sorry for our inaccurate expression. For the question of language, we invited the American journal expert to help us modify the description intensively. Hope the revised version will meet the criteria of publication, and thanks a lot for your comments.

We have corrected this phrase as follows: (Page 1 lines 26-28)

“…Furthermore, luteolin was also found to effectively relieve oxidative stress and promote autophagy in β cells, possibly improving β cell function and slowing the progression of diabetes.…”

  1. In addition, I take note that the authors do not provide higher resolution images of their WB results, as requested. I was not asking for better bands (they are what they are), but better images of said results.

Response: We feel sorry that the resolution of the images is not enough in the figure format and we could not improve them to higher resolution images. In our facility, the images of the WB were taken by Bio-Rad, then exported from the Image Lab software for application for figure arrangement. The current image is the highest resolution available. In case of future publication, we can e-mail the original images to editor for proofreading.

Round 3

Reviewer 3 Report

The authors presented a revised version of the paper I have previously reviewed.

The authors have appropriately addressed all my comments now.

Thus, the manuscript can be accepted in its present form.

Reviewer 4 Report

I have nothing to add since my last review. I don't even understand why I was sent this submission a third time.